# High Frequency of Antibiotic Resistance Genes (ARGs) in the Lerma River Basin, Mexico

**DOI:** 10.3390/ijerph192113988

**Published:** 2022-10-27

**Authors:** Ana K. Tapia-Arreola, Daniel A. Ruiz-Garcia, Hectorina Rodulfo, Ashutosh Sharma, Marcos De Donato

**Affiliations:** School of Engineering and Sciences, Campus Queretaro, Tecnologico de Monterrey, Santiago de Querétaro 76130, Querétaro, Mexico

**Keywords:** MDR, Lerma basin, ARGs, Chapala Lake, WGS

## Abstract

The spread of beta-lactamase-producing bacteria is of great concern and the environment has been found to be a main source of contamination. Herein, it was proposed to determine the frequency of antimicrobial-resistant-Gram-negative bacteria throughout the Lerma River basin using phenotypic and molecular methods. Resistant bacteria were isolated with chromogenic media and antimicrobial susceptibility tests were used to characterize their resistance. ARGs for beta-lactams, aminoglycosides, and quinolones were detected by PCR. Species were identified by Sanger sequencing the 16S rRNA gene and the representative genomes of MDR strains were sequenced by NGS. A high variation in the number of isolates was observed in the 20 sampled sites, while observing a low diversity among the resistant bacteria. Of the 12 identified bacterial groups, *C. freundii*, *E. coli*, and *S. marcescens* were more predominant. A high frequency of resistance to beta-lactams, quinolones, and aminoglycosides was evidenced, where the *bla*_CTX,_
*qnr*B, *qnr*S y, and *aac*(6′)lb-cr genes were the most prevalent. *C. freundii* showed the highest frequency of MDR strains. Whole genome sequencing revealed that *S. marcescens* and *K. pneumoniae* showed a high number of shared virulence and antimicrobial resistance genes, while *E. coli* showed the highest number of unique genes. The contamination of the Lerma River with MDR strains carrying various ARGs should raise awareness among environmental authorities to assess the risks and regulations regarding the optimal hygienic and sanitary conditions for this important river that supports economic activities in the different communities in Mexico.

## 1. Introduction

Antimicrobial resistance (AMR) can be found among bacteria that inhabit pristine environments and is a response to antimicrobials produced by the members of the communities in those environments [1,2]. This is evidenced by the presence of antibiotic resistance genes (ARGs) in permafrost, forest soils, ocean beds, mountains, and polar regions that have been identified from antimicrobial-resistant bacteria (ARB) [3]. However, the widespread use of antibiotics, among anthropogenic environmental changes around the world, has been associated with the evolution and prevalence of antimicrobial-resistant pathogens, as well as ARGs; consequently, this has created a global health concern [2]. The increase, evolution, and dissemination of multidrug resistance (MDR) depends on the microorganism, the host organism (human or other animal), the environments involved, and cultural and socioeconomic characteristics. The spread and permanence of ARGs also depend on their integration into mobile genetic elements (MGEs) and interaction networks between ecologically connected populations of bacteria [4,5,6,7,8,9]. Conjugative plasmids related to the F factor are highly associated with virulence and antibiotic resistance among the Enterobacteriaceae, and the wide spread of this group of plasmids carrying ARGs has been linked with human activities [10,11].

The potential ecological problem of antibiotic contamination affects waterways, and several studies point to waste products as potential enhancers that capture the consequences of human activities and have the potential for extensive environmental contamination. Rivers and other watercourses have been studied for their importance as reservoirs of resistant bacteria and carriers of ARGs. Studies of urban, medical, and industrial wastewater treatment plants have identified a positive correlation between the presence of antibiotics and bacteria resistant to them [12,13,14,15].

For the treatment of infections caused by Gram-negative bacteria, fluoroquinolones, cephalosporins, and combinations of β-lactam inhibitors are the most widely used antimicrobials due to their efficacy, safety, and availability. However, the TEM and SHV-type extended spectrum beta-lactamase-producing (ESBL) Gram-negative bacteria are resistant to oxyimino cephalosporins, while CTX-type enzymes preferentially hydrolyze cefotaxime over ceftazidime. ESBL-producing bacteria commonly present cross-resistance to other classes of antimicrobials (gentamicin, ciprofloxacin, and trimethoprim-sulfamethoxazole, among others) and are mobilized in MGEs that allow for the greater expression of these genes and confer a clinically relevant level of resistance [16,17].

In a longitudinal study of various sources of wastewater in France, CTX-M type, ESBL-producing *E. coli* occurred in most samples at significantly higher levels in hospital wastewater than in community wastewater. This result likely reflects the higher rates of carriers of ESBL among hospitalized patients compared to community inhabitants, and the difference in the levels of selective pressure of the residual antibiotics in the wastewater [18,19].

Another potential environmental source for ESBL producers is wildlife, especially birds. Studies of wild bird species such as gulls, aquatic birds, and waders show different frequencies of *E. coli* producing *bla*_CTX-M_, *bla*_SHV,_ and *bla*_TEM_-type ESBLs presented in fecal samples, thus showing birds’ ability to spread ESBLs genes, which can freely migrate between urban areas and agricultural lands, playing a role in the spread of resistance in different geographic areas and ecological niches [20,21,22].

The Lerma River is a very important source of freshwater for west-central Mexico. At 965 km long, it begins at an altitude of more than 3000 m above sea level in the Mexican plateau, 24 km southeast of the city of Toluca (Estado de Mexico), ending at Lake Chapala, in the state of Jalisco [23]. Throughout the basin, agricultural activities are developed with respect to 750,000 ha of irrigation, livestock activities, more than 1500 industries, and important urban centers, which contribute to the release of waste, converting this basin into the most polluted region of the country [24]. The rapid dissemination of ESBL-producing bacteria is likely multifactorial, but comprehensive knowledge of the environmental contamination in this basin is necessary. Since no previous reports of a cross-sectional study in the Lerma River basin have been published, the present study aims to determine the frequency of Gram-negative bacteria resistant to antimicrobials along the Lerma River basin using phenotypic and molecular methods, as a way to establish a baseline dataset from which future, more detailed and specific studies could be designed.

## 2. Materials and Methods

### 2.1. Study Sites and Sample Collection

A cross-sectional study of 20 sites along the Lerma River basin, as well as Lake Chapala, was carried out to evaluate the presence of antimicrobial resistance genes. Fourteen of the sites were artificial dams or natural lakes and the rest were established directly in the Lerma River. All the samples were taken in February 2021, when the daily average environmental temperatures for these states were 14–18 °C and the precipitation was the lowest in the year, with no major rain events in any of the sites 30 days before the sampling. In each site, two 250 mL samples were obtained for further processing. The samples were concentrated by centrifugation and enriched in LB (Luria Bertani) broth at 37 °C, and after 24 h, they were plated onto CHROMagar^TM^ ESBL and CHROMagar^TM^ mSuperCARBA to identify ß-lactam resistance bacteria.

### 2.2. Antimicrobial Susceptibility

The agar diffusion method was used to evaluate resistance to the following antimicrobials: ceftazidime (30 µg), ceftazidime/clavulanic acid (30/10 μg), cefotaxime (30 µg), cefotaxime/clavulanic acid (30/10 μg), imipenem (10 µg), meropenem (10 µg), ciprofloxacin (5 µg), levofloxacin (5 µg), tobramycin (5 µg), netilmicin (30 µg), and gentamicin (10 µg). All assays were conducted using the criteria established by Clinical and Laboratory Standards Institute as sensitive, intermediate, and resistant, using the specified tables for Enterobacteriaceae, *Pseudomonas aeruginosa,* and *Acinetobacter* spp. The quality control of the antimicrobial disks was verified by using the control strains *Escherichia coli* (ATCC 25922) y *Pseudomonas aeruginosa* (ATCC 27853).

### 2.3. Molecular Detection of Bacterial Species

DNA from each isolate was extracted using the Wizard Genomic DNA Extraction Kit (Promega), following the instructions from the manufacturer. For the amplification of the 16S rRNA gene, the bacterial universal primers used were: fD1-F: 5′- AGAGTTTGATCCTGGCTCAG-3′ and rP2-R: 5′- ACGGCTACCTTGTTACGACTT-3′ [25]; they generated an amplified product of about 1500 bp. The PCR products were purified and sequenced by the Sanger protocol at the Laboratorio de Servicios Genomicos del Laboratorio Nacional de Genomica para la Biodiversidad (LANGEBIO), Cinvestav, and the results were evaluated by the comparison of the 16S rRNA gene sequences from reference genomes published at GenBank NCBI (National Center for Biotechnology Information) in order to identify the strains. A species was assigned only when the sequence showed an identity of 99.0% or higher to a single reference species.

### 2.4. Detection of Resistance Genes

The detection of resistance genes was performed for the 59 strains to identify resistance to the following antimicrobial categories: ß-lactams, carbapenems, fluoroquinolones, and aminoglycosides. The genes *bla*_CTX_, *bla*_TEM_, *bla*_SHV_, *bla*_OXA_, *bla*_NDM_, *bla*_KPC_, *bla*_VIM_, *qnr*A, *qnr*B, *qnr*S, and *aac*(6′)Ib-cr were used to screen all isolates by PCR, using corresponding primers (Appendix A) to identify sources of antimicrobial resistance. All the PCR products were evaluated in 2% agarose gels in TBE 1X buffer at 80 V for 60 min. To compare the fragment sizes of the amplified DNA, a molecular weight marker was used: 100 bp Invitrogen (ThermoFisher Scientific). The DNA was stained by including GelGreen dye in the gel. The amplified products were detected by observation in an iBright CL1000 (ThermoFisher Scientific) and digitally recorded.

### 2.5. Whole Genome Sequencing

Selected strains from the species *E. coli*, *C. freundii,* and *K. pneumoniae* showing a multidrug resistance pattern—isolated from sites at the beginning and the end of the basin—were selected for whole genome sequencing, which was carried out at the Bioengineering Center of Tecnologico de Monterrey, Campus Querétaro, using the Nextseq 550 Illumina platform. The libraries were generated with the Nextera DNA Flex library preparation kit (Illumina^®^, San Diego, CA, USA). The size and quantity of the libraries generated were analyzed using the Fragment Analyzer employing the HS NGS Fragment Analysis Kit (Agilent Technologies, Santa Clara, CA, USA). The sequencing was carried out using V2 chemistry for 2 × 150 cycles in a Mid Output flow cell, in accordance with the standard Illumina-sequencing protocol. The reads were demultiplexed with bcl2fastq2 (Illumina^®^).

### 2.6. Bioinformatic Analysis

The quality of reads was evaluated by FASTQC software (http://www.bioinformatics.babraham.ac.uk/projects/fastqc/, accessed on 1 October 2021). Subsequently, the low-quality reads were trimmed and removed with Trimmomatic 0.36.0 [26], using the following parameters: an average quality per base of Q30, sliding window of 4 pb, and minimum length of 200 pb. Then, the trimmed reads were assembled into contigs using SPAdes Genome Assembler version 3.9.0 [27]. For the genome annotation, the Prokka Genome Annotation software tool was used [28]. Bacterial Analysis Pipeline version 1.0.4 [29] was used to identify the strain species, antimicrobial resistance genes, and to perform multilocus sequence analysis (MLSA). Additionally, virulence genes were identified by searches through BLAST and analysis of the virulence factor of pathogenic bacteria database (VFDB) [30].

## 3. Results

### 3.1. Bacterial Distribution and Characterization

In total, 20 water sources among five states of Mexico (Estado de Mexico, Queretaro, Michoacan, Guanajuato, and Jalisco) were sampled from the upper to lower course of the Lerma River basin, obtaining 59 environmental isolates identified as ß-lactam-resistant-Gram-negative bacteria. The details of the frequency of the isolated strains are shown in Figure 1 according to the water source from which they were obtained, revealing that the highest frequency of Gram-negative bacteria isolates corresponds to the Solis Dam (site 12) in Guanajuato with 10.2%, followed by San Nicolas and Chapala Centro (site 19 and 20 respectively) in Jalisco, which present a frequency of 8.5% each. In addition, we observed a high variation in the number of isolates along the 20 sites that were sampled, as well as low bacterial diversity among the resistant bacteria, in the specific sites where only one strain was obtained (Figure 1 and Figure 2).

From the 16S rRNA sequencing of the Gram-negative bacteria isolates, 12 different bacterial groups were identified: *Citrobacter freundii*, *Escherichia coli,* and *Serratia marcescens* (representing 15.3% of all isolates); *Pseudomonas putida* (13.6%); *Klebsiella pneumoniae* (11.9%); *Pseudomonas otitidis*, *Pseudomonas* sp., and *Acinetobacter pittii* (6.8% each); *Raoultella ornithinolytica* (3.4%); and *Escherichia fergusonii*, *Pseudomonas alcaligenes,* and *Pseudomonas sihuiensis* (1.7% each). The geographical distribution of the resistant isolates showed that some water sources presented a higher diversity of species than others (Figure 2). Cienegas (site 3), Solis Dam (site 12), and San Nicolas (site 19) presented four species, whereas in sites 4, 8, 13, 14, and 18, only one species was identified as ß-lactam-resistant-Gram-negative bacteria in each of the sites.

### 3.2. Antimicrobial Resistance in the Lerma River

In the antimicrobial susceptibility tests, nine antimicrobials were evaluated in the 59 ß-lactam-resistant-Gram-negative bacteria isolates, including the ß-lactams cephalosporins and carbapenems, aminoglycosides, and quinolones. As expected, the Lerma River basin presented higher antimicrobial resistance against the ß-lactam category (Figure 3A), showing resistance to cefotaxime (CTX) as the highest of all antimicrobials, representing 58 of the 59 isolates, followed by ceftazidime (CAZ), with a frequency of resistance of 59.3%. In addition, the carbapenems imipenem (IMP) and meropenem (MEM) presented frequencies above 44%. The molecular detection of ARGs revealed the presence of the gene *bla*_CTX-M_ in 88% of all the evaluated strains, placing it as the most abundant ARG (Figure 3B) in the Lerma River basin that can confer resistance to a wide range of ß-lactams. Other ß-lactam resistance genes such as *bla*_SHV_, *bla*_OXA_, *bla*_TEM_, and *bla*_VIM_ were less frequent, showing frequencies of 10%, 7%, 5%, and 3%, respectively; however, these are not less important, since *bla*_OXA_ and *bla*_VIM_ confer resistance to carbapenems.

Furthermore, the second most frequent antimicrobial category was represented by fluoroquinolones (Figure 3A), with ciprofloxacin (CIP) and levofloxacin (LVX) presenting resistances of 67.80% and 50.85%, respectively. The ARGs for fluoroquinolone resistance were, consequently, the second most frequent in the strains, with *qnr*B and *qnr*S showing 51% and 14% frequencies, respectively (Figure 3B). Finally, resistance to aminoglycosides was the least frequent of all the antimicrobial resistances studied, wherein antibiotic tobramycin (NN) had a resistance of 44.07%, followed by gentamicin (GM) and netilmicin (NET) with resistances of 11.86% and 3.60%, respectively. The ARG *aac*(6′)lb-cr showed a frequency of 10% among the isolates.

### 3.3. Multidrug Resistance in the Lerma River Basin

The distributions of the antimicrobial categories in each of the 20 sites were studied based on the results of the antimicrobial susceptibility test. The cephalosporin category is present throughout the Lerma River, followed by carbapenems, which are present in 18 of the sites (Figure 4A). These two ß-lactam categories together correspond to ≥40% of the AMR in every site. Moreover, fluoroquinolone and aminoglycoside resistance were observed in 15 sites each, but with a higher predominance of fluoroquinolone resistance, and following the proportions previously described in Figure 3. Overall, 17 sites presented AMR to more than three antimicrobial categories and only three sites (6, 7 and 13) showed the presence of two antimicrobial categories (see Figure 4A).

The results obtained from the antimicrobial susceptibility test showed that from the 59 isolates identified of the 12 bacterial groups, only 9 bacterial groups presented a multidrug resistance phenotype (Figure 4B). The data showed that *Citrobacter freundii* was the species with the highest MDR frequency, namely, 13.56% of all isolates (8/59), followed by *Serratia marcescens* and *Klebsiella pneumoniae* with 11.86% and 10.17%, respectively. The lower frequencies were identified in *Escherichia fergusonii*, *Raoultella ornithinolytica*, and *Acinetobacter pittii* (1.69% each). In addition, *Pseudomonas alcaligenes*, *Pseudomonas otitidis,* and *Pseudomonas sihuiensis* were the only species from the order of Pseudomonadales with no MDR phenotypes (Figure 4B).

The MDR strains with their respective ARGs are shown in Table 1, where a color map of the species shows the distribution of MDR across the Lerma River basin. The table shows that all MDR strains presented resistance to at least three of the evaluated antimicrobial categories. Evaluating the presence of ARGs showed that all the strains were carriers of the *bla*_CTX-M_ gene except three strains from the San Fernando Dam and La Marimba Bridge (site 8: LR39-08CF LR58-14CF and LR60-14CF), while the *bla*_SHV_ gene was only present in Cienagas de Lerma, Cienagas, Acambaro, and San Nicolas (sites 2, 3, 15, and 19); the *bla*_TEM_ gene was only found in the strains from the Melchor Ocampo Dam and San Nicolas.

The genes conferring resistance to carbapenems were identified in the Amoloya River, Cienagas, La Marimba Bridge, and Acambaro (sites 1, 3, 14, and 15). Of the genes associated with resistance to aminoglycosides and quinolones, *aac*(6′)lb-cr and *qnr*S were detected in only a few strains, but *qnr*B was identified in 25 of the 36 MDR strains. Finally, two *K. pneumoniae* strains from site 19 were the only species non-susceptible to the nine antimicrobials used in this study, carrying ARGs against cephalosporins, aminoglycosides, and fluoroquinolones.

Despite the limitations of this being a preliminary study, we carried out a binary logistic regression analysis to detect differences in the presence of resistance to antimicrobials, ARGs, and MDR, which compared states, regions, and sampling types. A binary logistic regression concerning the frequency of MDR strains according to the type of sampling site showed a higher frequency in river samples than in lakes (*p* = 0.047), while dams had an intermediate frequency between the other two types. This could be explained by the fact that larger water bodies have a more complex microbial ecosystem, which is less prone to be dominated by MDR strains.

The analysis of the whole genome sequencing of four *E. coli*, two *C. freundii,* and one *K. pneumoniae* isolate(s) allowed for the detection of ARGs conferring resistance to other aminoglycosides (*aac*(3)-Iia, *aac*(3)-IId, and *aad*A1), other types of beta-lactams (*amp*H, *bla*_CMY-70_, *bla*_DHA-1_, and *cph*A1), other quinolones (*oqx*A, *oqx*B), and to the sulphonamide (*sul*1 and *sul*2), rifampicin (ARR-3), phenicols (*cat*A1, *cat*B3, *cat*B4, and *cat*B8), trimethoprim (*dfr*A1, *dfr*A14, and *dfr*A17), forfomycin (*fos*A), macrolide (*mph*(A)), streptomycin (*str*A and *str*B), and tetracycline (*tet*(A), *tet*(B), and *tet*(C)) present in these strains (Appendix A). The presence of these genes was associated with the phenotypes of MDR and suggests that many of the strains carry multiple ARGs not detected by PCR. Although these data are from only seven WGS strains, the presence of multiple ARGs not detected by PCR shows a potentially large number of ARGs present in the environmental strains of the Lerma River basin.

From the seven genomes sequenced among the selected strains, we were able to identify only two strains of *E. coli*: ST3541 and ST940. This is not uncommon when working with environmental strains, since the great majority of the sequence-typed clones come from clinical isolates.

Additionally, we were also able to identify different virulence genes, including those coding for fimbriae (*lpf*A, *fim*A, and *fim*H), glutamate decarboxylase (*gad*), a hexosyltransferase homolog (capU), type IV pilin (ppdD), increased serum survival (*iss*), escape from the phagosome (*dsb*A), and invasion (*ibe*B, *yij*P, *rel*A, *asl*A, *omp*A, and *dam*). Other genes detected were related to the formation of biofilms, capsule formation, immune evasion, and the secretory system, among others. This suggests that many of these enterobacteria can cause infections in human and animal hosts of clinical importance. The detection of antimicrobial and virulence genes in the genomes of the species sequenced showed a greater number of unique genes in *E. coli* compared to *C. freundii* and *K. pneumoniae*. In fact, no unique genes of either class were detected in *C. freundii*, but many were shared with *E. coli* (Figure 5).

## 4. Discussion

Due to the indiscriminate use and release of antimicrobials in the environment, rivers and lakes can act as reservoirs with the right conditions for the development of ARBs and ARGs in the water, affecting microorganisms and creating a public health problem [31,32,33,34]. This can be clearly evidenced in the Lerma River, which is highly susceptible to contamination by residual antimicrobials that are released through different sources, such as agricultural runoff, wastewater discharges, and leaching from nearby farms throughout the basin and where Gram-negative bacteria resistant to ß-lactams, aminoglycosides, and quinolones were detected throughout the entire basin, finding the highest frequency of resistance in the beginning, middle, and end of the basin.

The environmental diffusion of ARBs harboring ARGs has been reported in freshwater environments in different countries in Europe, Asia, and the Americas [35,36,37,38,39,40,41,42], including Mexico [43,44,45,46]. However, several factors such as geographic location and antimicrobial prescription policies can influence the proportion of ARBs in these environments [47].

The wide variety of Gram-negative bacteria species resistant to ß-lactams found—such as *Citrobacter freundii*, *Serratia marcescens*, *Klebsiella pneumoniae*, *Acinetobacter pittii*, *Raoultella ornithinolytica*, and different species of *Escherichia* and *Pseudomonas*—shows that contamination with these strains can lead to a higher risk of infections in humans and animals, and they are a public health problem because ß-lactamase-producing enterobacteria can cause epidemic infections [48,49].

ß-lactam-producing Enterobacteriaceae such as *E. coli*, *K. pneumoniae*, *E. cloacae*, *C. frendii*, *Enterobacter asburiae,* and *K. oxytoca* have been previously reported in streams and rivers [37,50,51], which can be coupled with the identification of species from the genera *Aeromonas*, *Pseudomonas,* and *Acinetobacter* carrying ß-lactamases, thus presenting a cause for concern due to their potential to act as vehicles for the dissemination of different mechanisms of resistance to human pathogens, especially since many of the ARGs present in these bacteria are associated with conjugative plasmids related to the F factor [52,53,54,55,56,57].

Higher levels of ARBs have been reported by Nnadozie and Odume [49] in river (98%) and lake (77%) systems compared to dams, ponds, and springs (<1%). In addition, natural dilution and degradation processes do not completely eradicate ARGs and ARBs. Although they are effectively inactivated by sunlight, their DNA (which contains ARGs) may be intact, and they can transfer resistance to susceptible strains. The direct transmission of indigenous freshwater ARBs to humans, as well as their transient insertion into the microbiota, may also occur.

In the Lerma River basin, *bla*_CTX-M_ was widespread in the isolated strains throughout the river basin. The detection of the *bla*_SHV_, *bla*_TEM_, *bla*_OXA_, and *bla*_VIM_ genes was also epidemiologically important, although such genes were found in few sites. However, the detection of ESBL-carrying-Gram-negative bacteria in the Lerma River is epidemiologically of great importance because these enzymes are encoded in plasmids, facilitating their spread to other bacterial strains and species, and their enzyme variability allow bacteria to hydrolyze different categories of ß-lactams [58,59,60]. Carbapenemase-producing bacteria represent a greater challenge to public health [51], as these antimicrobials are used as the last resource for treating dangerous infections caused by ESBL-producing strains [61]. These carbapenemase-producing bacteria have been designated urgent threats associated with high mortality rates [62,63]. These carbapenemases are also encoded by plasmids or integrons [37,60].

Another key aspect regarding the resistant strains from the Lerma River is that fluoroquinolones were the second most frequent category of antimicrobial resistance, demonstrated in CIP and LVX and through the *qnr*B and *qnr*S genes. One possible explanation is that fluoroquinolones are more stable than other antibiotics and remain in the environment longer. In river environments around the world, quinolones have been found in significantly higher concentrations [64]. In the case of aminoglycosides, they were the least frequent, but the *aac*(6′)lb-cr gene associated with resistance to quinolones and aminoglycosides was present in 10% of the resistant strains evaluated.

The frequency of MDR in the species identified in the Lerma River showed that *C. freundii* was the species with the highest MDR, followed by *Serratia marcescens*, *K. pneumoniae,* and *E. coli*. The most affected areas by these MDR strains were the Amoloya River, Ciénegas, La Marimba Bridge, Acámbaro, and San Nicolás, possibly due to the environment being more influenced by agricultural, aquacultural, and industrial activities. In general, the variations in the bacterial groups identified as MDR in freshwater may be due to the difference in the availability of the waste treatment system in hospitals, industries, farms, and other wastewater sources, as well as regional differences with respect to the use of antimicrobials [65].

The success of clones thriving in aquatic habitats is of particular interest for assessing the impact of each individual clone in order to prevent the emergence and spread of MDR clones. In the Lerma River, these types of strains with different virulence factors and ARGs were identified by sequencing. The *E. coli* ST3541 identified herein was also reported in a river in Algeria, which was affected by contamination from agricultural, industrial, and domestic sources [66]. This MDR strain carried a carbapenemase type, *bla*_OXA-48_ (variant 244), as did the one isolated in this study from Río Lerma with variant type 181 of *bla*_OXA-48_. The other strain identified in the Lerma basin was *E. coli* ST940, which has been reported as MDR in patients from the US, India, and Ghana [67,68,69], carrying different ARGs including *bla*_TEM_, *bla*_CTX-M_, *bla*_OXA_, and metallo-ß-lactamase (NDM) [67]. Bacterial clones are subjected to continuous evolution of the genome, such that the resistant strains identified in the Lerma River could have acquired greater resistance through mobile genetic elements as a faster and more extensive process of the diversification of strains within clonal complexes [70].

The strains sequenced here showed associated plasmids, which constitute the most important mechanism of ARG dissemination in the environment among different species. These ARGs are considered contaminants of emerging concern because some of the genes are present in bacteria as innate reservoirs. Specifically, IncF plasmids have been associated with multidrug resistance, usually carrying *bla*_CTX-M_ with other resistance genes such as Plasmid-Mediated Quinolone Resistance (PMQR) genes. Five of the seven sequenced strains from the Lerma River contain at least one IncF plasmid, mainly in *E. coli*, followed by *C. freundii* and *K. pneumoniae*. Furthermore, IncFII plasmids have been reported in isolates from wastewater treatment plants with high homology to clinical isolates, suggesting that ARGs are strongly linked with anthropogenic activities [11,71,72].

According to Fall et al. [73], a part of the upper Lerma basin presents some variables evaluated outside the limits accepted by Mexican Official Standard NOM 001, and most studies on the occurrence and fate of ARB and ARG in streams, lakes, and rivers are mainly focused on the discharge points of wastewater treatment plants; however, even when the concentration of antibiotics in the water is low, a selection of resistant bacteria can occur [33,74]. Additionally, the reported variations in the abundance of ARGs have been little studied, but could be due to several factors, including the sampling period and seasons, as well as the available methods [75]. The strains and the resistance genes reported so far in freshwater bodies do not represent the full range of ARGs since there is not a sufficient amount of research studying these ecosystems. The abundance of other resistance genes may not be sufficient to allow for them to be captured by the detection methods applied in each study. Whole genome sequencing of ARB can further illustrate the ARGs found in the environment and could be a very important tool for studying impacted environments such as the Lerma River basin.

## 5. Conclusions

To the authors’ knowledge, this is the first study of the frequency of antimicrobial-resistant-Gram-negative bacteria throughout the entire Lerma River basin. Herein, it has been shown that ß-lactamase-producing ARBs were found throughout the basin, highlighting the important frequency of MDR strains and the presence of many ARGs that represent contaminants of importance for environmental and public health. However, further studies are required to identify the sources of antimicrobial contamination. This research should raise awareness among health authorities and encourage them to review antimicrobial use regulations in the region and better monitor antimicrobial contamination levels to avoid the health risks that ARBs pose to communities.

## Figures and Tables

**Figure 1 ijerph-19-13988-f001:**
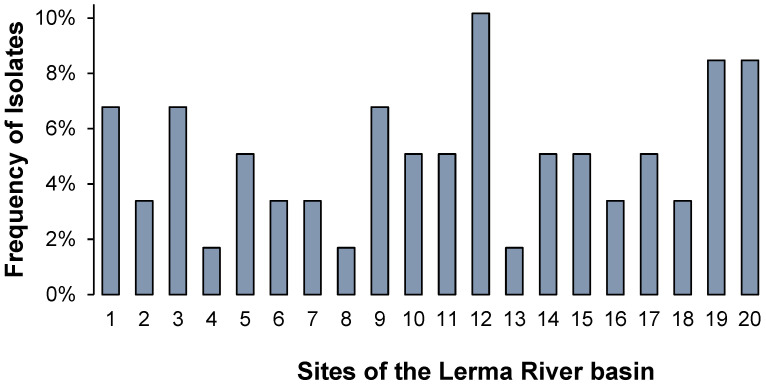
Frequency (%) of ß-lactam-resistant-Gram-negative bacteria isolated from the Lerma River basin according to the sampling location (1 through 20).

**Figure 2 ijerph-19-13988-f002:**
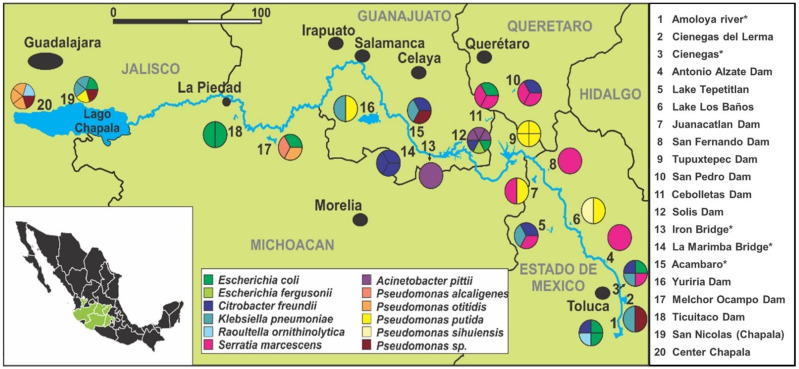
Geographical distribution of bacterial species identified along the five states that are part of the Lerma River basin. The diversity and number of species isolated from each site is represented in circle charts. In the sites with asterisks, samples were taken directly from the Lerma River.

**Figure 3 ijerph-19-13988-f003:**
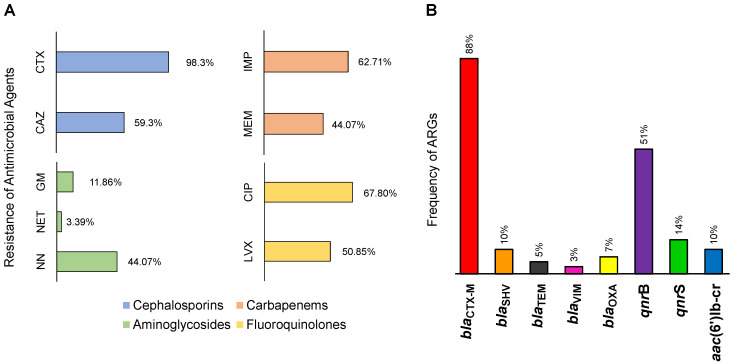
Frequency of resistance to antimicrobial agents and of ARGs that were detected in the Lerma River basin. (**A**) Antimicrobial agents. CAZ: ceftazidime, CTX: cefotaxime, IMP: imipenem, MEM: meropenem, NET: netilmicin, NN: tobramycin, GM: gentamicin, LVX: levofloxacin, and CIP: ciprofloxacin. (**B**) Frequency of ARGs that were studied and detected in the studied strains.

**Figure 4 ijerph-19-13988-f004:**
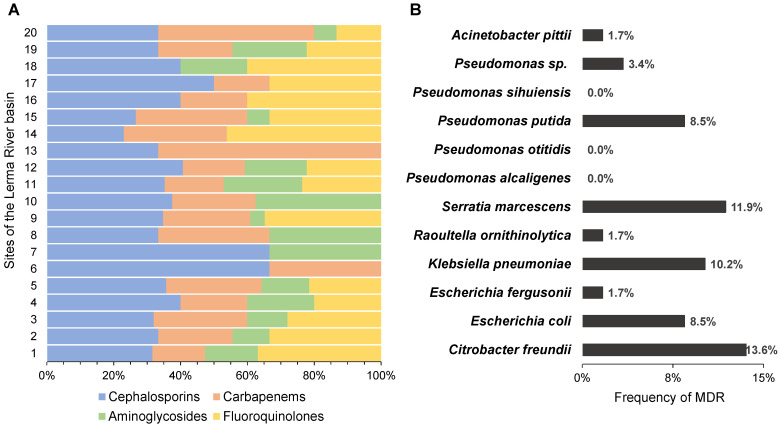
Antimicrobial resistance and MDR in the Lerma River basin. (**A**) Graphical distribution of the studied antimicrobial categories; each number represents locations of the Lerma River (see Figure 5 for details). (**B**) Frequency of environmental MDR strains, by species, found in the Lerma River.

**Figure 5 ijerph-19-13988-f005:**
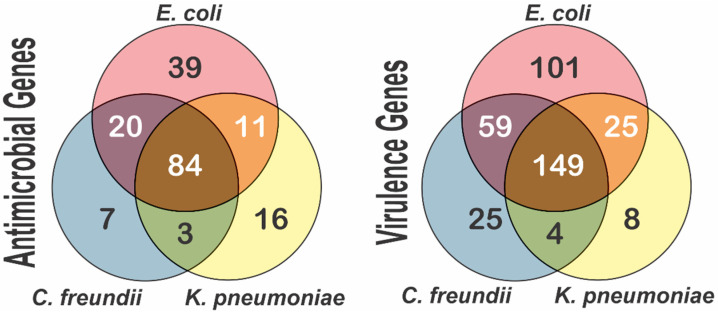
Venn diagram of the antimicrobial resistance and virulence genes identified in the genomes of the strains whose genomes were sequenced.

**Table 1 ijerph-19-13988-t001:** Multidrug resistance strains and resistance genes identified among the isolates. The strains in red were the ones selected for WGS.

	Site	1			2		3				4	5		8	9				10		11			12			14		15			16		17	18	19		20
	Strain	LR12-01CF	LR13-01EC	LR14-01EC	LR16-02KP	LR17-02PSP	LR18-03CF	LR25-03EC	LR26-03KP	LR27-03SM	LR28-04SM	LR29-05CF	LR31-05SM	LR39-08SM	LR40-09PP	LR41-09PP	LR42-09PP	LR43-09PP	LR44-10CF	LR46-10SM	LR47-11EC	LR48-11SM	LR49-11SM	LR51-12AP	LR54-12CF	LR56-12EF	LR58-14CF	LR60-14CF	LR33-15CF	LR61-15KP	LR62-15PSP	LR63-16KP	LR64-16PP	LR34-17EC	LR35-18EC	LR69-19KP	LR53-19KP	LR76-20RO
Antimicrobial category	Agent																																					
β-lactams	ceftazidime																																					
	cefotaxime																																					
carbapenems	meropenem																																					
	imipenem																																					
aminoglycosides	trobamycin																																					
	netilmicin																																					
	gentamicine																																					
fluoroquinolones	levofloxacin																																					
	ciprofloxacin																																					
	ARGs																																					
β-lactams	blaCTX-M																																					
	blaSHV																																					
	blaTEM																																					
carbapenems	blaOXA																																					
	blaVIM																																					
aminoglycosides	aac(6′)-lb-cr																																					
fluoroquinolones	qnrB																																					
	qnrS																																					
			*C. freundii*						*E. coli*						*S. marcescens*							*K. pneumoniae*							*E. fergusonii*									
			*A. pittii*						*P. putida*						*Pseudomonas sp.*							*R. ornithinolytica*																

## Data Availability

Data are available from the corresponding author upon request. The whole-genome sequence of the MDR bacteria can be found in the GenBank (SAMN31419015- SAMN31419021).

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
