# Peer review of "High Frequency of Antibiotic Resistance Genes (ARGs) in the Lerma River Basin, Mexico"

_ijerph, 2022, doi:10.3390/ijerph192113988_

Round 1

Reviewer 1 Report

Abstract: 

Please do not use or avoid abbreviations in the abstract.

Introduction:

It should be clearly stated that resistance genes are usually located on plasmids which in case of the studied/sequenced strains are very often conjugative plasmids from the F-like family. So the problem here is that ARGs can move from clinical strains to environmental strains of E.coli, Citrobacter, Klebsielle and vice versa. There are many reports and papers that indicate the dominance of this plasmid family in isolates from different sources and enterobacterial hosts. Interestingly, in table S2, in four out of six isolates there may again be IncF (F-like) conjugative plasmids that carry ARGs and/or virulence genes. 

Instead of GNB as an abbreviation I suggest to use “Gram-negative bacteria” throughout the text.

Methods:

A big drawback of the current study is that the NGS Genome sequences were not deposited in a public database. This should not be a too complex task (e.g. with NCBI and the creation of a BioProject)

Results:

Due to the rather low number of primary isolates one could just give absolute numbers instead of % values. At least I would like to see a table (supplement) in which each isolate is listed with the respective sampling site, resistance phenotype, bacterial identification, markers present etc.

In table 1 the mdr isolates that were chosen for WGS should be highlighted.

A presentation of what was the outcome of the genome sequencing effort is lacking although it is mentioned in the abstract.

Discussion:

The discussion could be considerably shortened without losing too much information. Again, what is above mentioned for the introduction (namely on plasmids as the main carriers of ABR genes), in my opinion, is also relevant for the discussion section. 

Author Response

REVIEWER 1

REVIEWER: It should be clearly stated that resistance genes are usually located on plasmids which in case of the studied/sequenced strains are very often conjugative plasmids from the F-like family. So, the problem here is that ARGs can move from clinical strains to environmental strains of E. coli, Citrobacter, Klebsielle and vice versa. There are many reports and papers that indicate the dominance of this plasmid family in isolates from different sources and enterobacterial hosts. Interestingly, in table S2, in four out of six isolates there may again be IncF (F-like) conjugative plasmids that carry ARGs and/or virulence genes.

ANSWER: We have addressed this concern and have added text in the introduction to include this information.

REVIEWER: Instead of GNB as an abbreviation I suggest using “Gram-negative bacteria” throughout the text.

ANSWER: DONE

REVIEWER: A big drawback of the current study is that the NGS Genome sequences were not deposited in a public database. This should not be a too complex task (e.g. with NCBI and the creation of a BioProject)

ANSWER: This is important suggestion, and we have deposited the sequences in the GenBank, and we added the temporal accession numbers. The final version of the manuscript will have the final accession numbers in the Data Availability Statement.

REVIEWER: Due to the rather low number of primary isolates one could just give absolute numbers instead of % values. At least I would like to see a table (supplement) in which each isolate is listed with the respective sampling site, resistance phenotype, bacterial identification, markers present etc.

ANSWER: We are submitting a supplementary table (S3) with the information of the 59 isolates characterized in this study.

REVIEWER: In table 1 the MDR isolates that were chosen for WGS should be highlighted.

ANSWER: We have highlighted the strains that we have WGS data in table 1.

REVIEWER: A presentation of what was the outcome of the genome sequencing effort is lacking although it is mentioned in the abstract.

ANSWER: In the last three paragraphs of the results, we added the results of the findings from the WGS analysis, and we added more details in the supplementary table S2.

REVIEWER: The discussion could be considerably shortened without losing too much information. Again, what is above mentioned for the introduction (namely on plasmids as the main carriers of ABR genes), in my opinion, is also relevant for the discussion section.

ANSWER: We have significantly modified the discussion to consider the comments of the reviewer. We have tried to shorten it but at the same time adding the information on the plasmids being the main carriers of ABR genes.

Reviewer 2 Report

The authors presented a study and manuscript regarding the contamination of Lerma River Basin in Mexico. The study was carried out with approximately 40 samples, 2 x 20 sites, presumably from a one time sampling regime. The authors then isolated various GNB on selective media and conducted phenotypic and some molecular analysis of these isolates. Overall, the study is very qualitative in nature and context from a quantitative standpoint would have improved the findings. The manuscript is well written apart from some areas which require more editing. Regarding the study, there needs to be more information regarding the sampling scheme, was this one time, and during what part of the year, what type of rain events preceded the sampling, etc.? Additionally, what criteria was used to select isolates for WGS. I assume that ARG assays were conducted on all isolates? The fact that the authors worked with 56 isoaltes, does this indicate that was all that could be detected in the samples, following enrichment in broth and plating, or were these randomly selected? Additionally, the authors should caution their interpretation of the results given the very qualitative nature of their sampling and assay regime. If I'm reading this correctly, the author enriched samples and only came away with 56 isolates, but without understanding the overall quantities associated with these isolates, we should caution our interpretations if these just represent a minimal portion of the microbiome or the river basin GNB population. That being said, it's important to document these results from countries like Mexico, and as such these results would be of interest to a greater public. 

Author Response

REVIEWER 2

REVIEWER: The study was carried out with approximately 40 samples, 2 x 20 sites, presumably from a onetime sampling regime.

ANSWER: Yes, this was a cross-sectional exploratory study to establish a base line dataset from which future, more detailed and specific studies could be designed. The samples were taken in a space of 30 days, sampling one to three sites per day.

REVIEWER: Overall, the study is very qualitative in nature and context from a quantitative standpoint would have improved the findings.

ANSWER: Yes, but since there was no base data, it is difficult to design the study without knowing what we could find. For example, our findings have shown that future studies must be carried out by genomic sequencing analysis in order to better identify the ARGs and the species carrying them.

REVIEWER: The manuscript is well written apart from some areas which require more editing.

ANSWER: We appreciate the effort of the reviewer to improve the manuscript. We have revised the whole manuscript and corrected some minor mistakes.

REVIEWER: Regarding the study, there needs to be more information regarding the sampling scheme, was this one time, and during what part of the year, what type of rain events preceded the sampling, etc.?

Answer: We have added the information requested in the materials and methods section.

REVIEWER: Additionally, what criteria was used to select isolates for WGS.

ANSWER: As it is stated in the materials and methods section, we selected 7 representative strains along the Lerma River basin that showed MDR pattern, which were classified as E. coli, C. freundii and K. pneumoniae, which are the most common clinical pathogens among the species found. These showed different molecular patterns for ERIC-PCR. The text was modified to make it more clear to the readers.

REVIEWER: I assume that ARG assays were conducted on all isolates?

ANSWER: YES. The text was modified to make it more clear to the readers.

REVIEWER: The fact that the authors worked with 56 isolates, does this indicate that was all that could be detected in the samples, following enrichment in broth and plating, or were these randomly selected?

ANSWER: After enrichment and growing the samples in selective media, we found 59 isolates. These were strains showing an ESBL resistance pattern and/or carbapenem resistance pattern. The non-resistant strains or those showing resistance only to other types of antimicrobials were not targeted. The text was modified to make it more clear to the readers.

REVIEWER: Was there any statistical analysis conducted on these samples? Comparisons between the sites, for instance?

ANSWER: As suggested by the reviewer, we carried out statistical analysis for the presence of resistance to antimicrobials, ARGs and MDR, among the states, the regions and the type of water body. We found only differences of the frequency of MDR between lakes and river samples, while no differences were found between lakes and dams or dams and river samples. This was written in the results section.

REVIEWER: The authors should caution their interpretation of the results given the very qualitative nature of their sampling and assay regime. If I'm reading this correctly, the author enriched samples and only came away with 56 isolates, but without understanding the overall quantities associated with these isolates, we should caution our interpretations if these just represent a minimal portion of the microbiome or the river basin GNB population.

ANSWER: We totally agree with the reviewer that this is a preliminary study to establish the baseline of future studies and that the results should be taken with caution. We have added this statement in the conclusions: "Since this is a preliminary study with a qualitative approach, these results should be taken with caution, because it doesn’t show the full extent of the Gram-negative species and ARGs present in the whole Lerma River basin"

REVIEWER: That being said, it's important to document these results from countries like Mexico, and as such these results would be of interest to a greater public.

ANSWER: Yes, we do believe that, even though this is a preliminary study, the findings are important to show the extent of the problem and the possible impact the this can cause to human and animal health.

REVIEWER: Is this frequency related to the PCR detections? or related to WGS? If PCR, is this due to PCR bias of just the pure number of B lactam targeted genes vs fluoros?

ANSWER: This is the frequency of the PCR detected genes, since it was carried out in all of the 59 strains. The genes detected by WGS was carried out in only 7 strains.

REVIEWER: While this may be correct, you only tested 7 isolates with combined phenotype analysis and WGS, so we should caution with making these types of assumptions.

ANSWER: This is true. We added the following statement to make a point clearer but showing caution because of the small set of strains WGS. "Although this is data from only 7 WGS strains, the presence of multiple ARGs not detect-ed by PCR show a potentially large number of ARGs being present in the environmental strains of the Lerma River basin"